# The Pursuit of the “Inside” of the Amyloid Hypothesis—Is C99 a Promising Therapeutic Target for Alzheimer’s Disease?

**DOI:** 10.3390/cells12030454

**Published:** 2023-01-31

**Authors:** Nobumasa Takasugi, Masato Komai, Nanaka Kaneshiro, Atsuya Ikeda, Yuji Kamikubo, Takashi Uehara

**Affiliations:** 1Department of Medicinal Pharmacology, Graduate School of Medicine, Dentistry and Pharmaceutical Sciences, Okayama University, Okayama 700-8530, Japan; 2Department of Cellular and Molecular Pharmacology, Juntendo University Graduate School of Medicine, 2-1-1 Hongo Bunkyo-ku, Tokyo 113-8421, Japan; 3Division of Biomedical Sciences, School of Medicine, University of California, Riverside, CA 92521, USA; 4Center for RNA Biology and Medicine, University of California, Riverside, CA 92521, USA

**Keywords:** Alzheimer’s disease, amyloid-β, amyloid beta precursor protein, BACE1, C99, endolysosome, autolysosome, vesicular trafficking

## Abstract

Aducanumab, co-developed by Eisai (Japan) and Biogen (U.S.), has received Food and Drug Administration approval for treating Alzheimer’s disease (AD). In addition, its successor antibody, lecanemab, has been approved. These antibodies target the aggregated form of the small peptide, amyloid-β (Aβ), which accumulates in the patient brain. The “amyloid hypothesis” based therapy that places the aggregation and toxicity of Aβ at the center of the etiology is about to be realized. However, the effects of immunotherapy are still limited, suggesting the need to reconsider this hypothesis. Aβ is produced from a type-I transmembrane protein, Aβ precursor protein (APP). One of the APP metabolites, the 99-amino acids C-terminal fragment (C99, also called βCTF), is a direct precursor of Aβ and accumulates in the AD patient’s brain to demonstrate toxicity independent of Aβ. Conventional drug discovery strategies have focused on Aβ toxicity on the “outside” of the neuron, but C99 accumulation might explain the toxicity on the “inside” of the neuron, which was overlooked in the hypothesis. Furthermore, the common region of C99 and Aβ is a promising target for multifunctional AD drugs. This review aimed to outline the nature, metabolism, and impact of C99 on AD pathogenesis and discuss whether it could be a therapeutic target complementing the amyloid hypothesis.

## 1. Introduction

Alzheimer’s disease (AD) is characterized by progressive cognitive decline and various behavioral abnormalities [1,2]. The number of patients is increasing in developed countries. Therefore, preventing AD or developing AD therapy could benefit the patients and contribute to public welfare.

The formation of abnormal structures, such as extracellular senile plaques, intracellular neurofibrillary tangles (NFTs), and progressive neuronal loss, are typical pathological features observed in an AD brain [3]. Tau, a microtubule-binding protein, is highly phosphorylated and deposited as NFTs; the appearance of NFTs correlates with neuronal cell death and cognitive impairment and is presumed to contribute to the progression of AD pathology [4]. However, in terms of pathogenesis, a more vital link has been established in the accumulation and aggregation of a small peptide called amyloid beta (Aβ), the primary component of senile plaques. At the time of writing this review, about 30 years have passed since the “amyloid hypothesis” was proposed, which places Aβ, an aggregating neurotoxic peptide, at the center of the AD pathogenic mechanism [4,5]. This hypothesis has a solid pathological and genetic basis. Aβ begins to deposit in the patients’ brains more than a few decades before the onset of AD, and many mutations related to early-onset familial AD (FAD) have been reported in the genes involved in Aβ production.

Much of the drug development based on this hypothesis has failed, but Aβ immunotherapy can break this long-time stagnation. However, as discussed in the later chapters, its therapeutic effects are currently limited. We must expand further from the success of Aβ immunotherapy and consider the hidden aspects of the amyloid hypothesis (Figure 1).

Aβ is produced from a large type-I transmembrane protein, amyloid beta precursor protein (APP). One of the APP metabolites, its 99-aa C-terminal fragment (C99, also called βCTF), is a direct precursor of Aβ. Our group and other researchers have demonstrated the significance of C99 in AD pathogenesis [6,7]. C99 accumulates through the endolysosomal and autolysosomal pathways, which could complement the missing part of the amyloid hypothesis, as “intracellular” pathology. This review aimed to explain the nature of C99, its contribution to AD pathogenesis, and discuss its potential as a drug target. Also, we suggested that targeting C99 would reduce Aβ dependent toxicity and has the potential to develop multifunctional drugs.

## 2. Generation of C99 and Aβ

Since APP has multiple splicing variants [8], the cleavage sites of each secretase and mutations described below were based on the Aβ sequence for simplicity. The two major pathways for APP metabolism include three proteases, namely, α-, β-, and γ-secretase. The non-amyloidogenic pathway involves a stepwise cleavage by the α- and γ-secretase. α-secretase is a member of the ADAM (a disintegrin and metalloprotease) family and is mainly distributed on the cell surface [9]. α-secretase cleaves within the Aβ sequence of APP to produce soluble APPα (sAPPα) and C83 (αCTF). The second pathway, the amyloidogenic pathway, involves β- and γ-secretase. β-secretase (BACE1) is an aspartate protease localized mainly to the Golgi, trans-Golgi network (TGN), and endosomes [10,11] and cleaves two sites of APP, the β-site (D1) or β’-site (E11). The former produces sAPPβ’ and C89 (β’CTF), and the latter produces sAPPβ and C99 (βCTF). As a final step in both the pathways, γ-secretase, a four-protein complex formed by Presenilin1 or Presenilin2-Aph1-Nicastrin-Pen-2 [12,13], resides in the Golgi, TGN, and endosomes, cleaves the transmembrane domain within the lipid bilayer of C83 and C99 to produce P3 and Aβ, respectively (Figure 2). Although this review focuses on Aβ and C99, other metabolic mechanisms of APP, such as δ-secretase and η-secretase cleavage, and their contribution to AD pathogenesis have been investigated in detail [14,15].

Several mutations in APP suggest the significance of β1 cleavage, including a cleavage-promoting Swedish mutation (KM-2/-1NL) [16] and a cleavage-inhibiting and protective Icelandic mutation (A2T) in the vicinity of the β1 cleavage site [17] (Figure 3). From a sporadic AD viewpoint, BACE1 is stress-sensitive, and many stresses affect its expression and activity [18]. BACE1 activity increases with aging [19], particularly in the brain of patients with mild cognitive impairment (MCI), a precursory symptom of AD [20,21,22]. Activation of BACE1 increase C99 and Aβ and correlates with the amount of Aβ plaque and cognitive decline in MCI [22].

Several Aβ species have different C-terminal lengths due to fluctuations in the γ-secretase cleavage site [23,24]. Among them, the primary species are Aβ40 and Aβ42. Aβ42 is highly cohesive and forms soluble and neurotoxic oligomers and protofibrils and is deposited in senile plaques from early phases [25,26]. Mutations in Presenilin1 and Presenilin2, the active centers of γ-secretase, and in APP itself have been identified in early-onset AD [15,27], many of which increase the amount of total Aβ [16] or the ratio of Aβ42/Aβ40 [28,29,30].

Therefore, research has focused on the regulation of Aβ (42) production, metabolism and aggregation [31]. In particular, the inhibitors of β- and γ-secretase involved in the amyloidogenic pathway have been developed, but most have failed [32]. One of the reasons is presumed that both secretases have other substrates other than APP. For instance, γ-secretase cleaves Notch1 [33,34], which regulates tissue cell differentiation, and β-secretase cleaves Neuregulin1 (NRG1) [35,36,37], which regulates neuronal and oligodendrocyte differentiation, so the simple inhibition of these enzymes can cause adverse effects. Inhibition of γ-secretase activity has been associated with gastrointestinal [38], and inhibition of BACE1 activity with demyelination [36,37], neurological symptoms [39,40].

Aβ immunotherapy has the potential to break the stagnation of AD drug development. To address the current status of AD drug development, we briefly introduced Aβ immunotherapy in the next chapter.

## 3. Potential and Limitations of Aβ Immunotherapy

An Aβ immunotherapy developed by Eisai and Biogen, a Japanese, and U.S. pharmaceutical company, is now attempting to clarify the correctness of this hypothesis from a therapeutic perspective.

Aβ immunotherapy originated from a breakthrough report in 1999 that immunization with Aβ42 prevents the development of amyloid plaque formation, neuritic dystrophy, and astrogliosis in the AD model, PDAPP mice [41]. Since then, many Aβ antibodies have been examined for their therapeutic effects in the AD model mice, such as the reduction of amyloid plaques [42,43,44]. The PRIME phase 1b study in patients with prodromal to mild AD showed that one year of monthly administration of aducanumab, which targeted aggregated Aβ reduces brain Aβ levels in a dose- and time-dependent manner and significantly slows clinical decline measured by cognitive function endpoints. Two subsequent phase III trials (EMERGE and ENGAGE study) initially failed to show the therapeutic effects of aducanumab [45]. However, a subsequent reanalysis of a more extensive data set showed that the highest doses of aducanumab (10 mg/kg) reduced the decline in the primary endpoint, the CDR-SB in EMERGE study, and led to conditional approval by the FDA. Moreover, in patients with early AD, lecanemab, an anti-protofibril Aβ antibody, reduced brain amyloid levels and indicated moderately less decline in clinical measures of cognition and function than placebo in eighteen months [46].

However, the efficacy and cost-effectiveness of Aβ antibody therapy have also been questioned [47,48,49]. Although they will further improve in the future by considering the timing of therapeutic interventions, the biggest concern is that even Aβ immunotherapy has only a limited effect, such as slowing cognitive decline. Note that the amyloid hypothesis at present has the limitation of majorly targeting the toxicity of “extracellularly” accumulated Aβ.

## 4. C99 Accumulation in AD Pathology and Other Neurodegenerative Diseases

Recently, C99 has attracted attention as an “intracellular” pathology promoter in the neurons. C99 is normally produced at low levels, but when amounts are increased, such as by activation of BACE1, it is not readily degraded and accumulates in the brain. This type of property is important in the etiologic factors in neurodegenerative diseases that progress with aging. This chapter outlines C99 accumulation and the methods for its analysis in AD and other neurodegenerative diseases and their models.

AD patients, particularly FAD patients with mutations in APP or Presenilin 1, showed a significant accumulation of C99 [50]. Similar accumulation of C99 has been reported in AD cell models in which APP and Presenilin 1 mutations of FAD were artificially introduced into iPS cells, suggesting that this is a common AD pathology [51,52]. Many of these mutations increase the production ratio of Aβ42, seemingly contradicting the accumulation of the substrate, C99. However, the increase in Aβ42 can be understood as a change in γ-secretase cleavage site. APP mutations in the proximal site of γ-secretase cleavage enhance the Aβ42 generation rate, but some of them tend to increase the APP-CTF stability [30]. It was also reported that FAD mutations in Presenilin1 shift the cleavage site toward Aβ42 production due to reduced secretase activity [29], or impaired proteolysis in autolysosome [53]. The commonality shown by these lines of evidence is that C99 is once produced, less likely to be degraded. Furthermore, as will be discussed in later chapters, there is also a vicious cycle in which C99 accumulation impairs the endolysosomal and autolysosomal pathways, which impairs the metabolism of C99 itself. These findings suggest that the pathogenic mechanism of AD has a cross link to the accumulation of C99.

Would C99 accumulation be associated with pathology? As C99 contains the Aβ sequence, histological differentiation of C99 and Aβ is difficult. Applying β1 cleavage site-specific (antibodies detect both Aβ and C99), and APP C-terminal antibodies to the proximal ligation assay (PLA) enables the analysis of the precise localization of C99. Using this technique, it was observed that C99 accumulated in AD brains in regions susceptible to AD, such as CA1 and the dentate gyrus, and that this correlated with pathological conditions such as cognitive decline [54]. Similar results were obtained with standard histological analyses using model mice [55]. Thus, the accumulation of C99 occurs in AD model mice, but most AD model mice have the Swedish APP mutation, which promotes the production of C99 and makes its accumulation artificial. Furthermore, additional mutations that accelerate Aβ aggregation are added in many models, making it difficult to distinguish between C99 and Aβ aggregation toxicity. Efforts are being made to distinguish them, including using γ-secretase inhibitors, which increase C99 while suppressing Aβ production, and β-secretase inhibitors, which decrease both Aβ and C99. Additionally, immunostaining using a combination of specific antibodies, the PLA analysis [54], ELISA [56], and immunoblotting [57] are used to determine the C99 specific distribution and quantification. These analyses showed that C99 accumulation precedes Aβ deposition in model mice and mediates Aβ independent intracellular pathologies [55,56,57,58]. However, limitations of the analysis using standard AD model mice may need to be addressed. For C99-specific analysis, models such as the A7 mouse [57,59], in which the Aβ deposition is slow, and the model without the Swedish mutation [60] may be helpful.

Additionally, other neurodegenerative diseases accumulate C99 and affect pathology, and these conditions can also provide important clues. Down syndrome (DS) is caused by trisomy of chromosome 21 and develops AD at a high rate and early age, mainly due to the triplication of the APP gene. Therefore, the DS can be a pathological model of AD. As discussed in later chapters, the initial pathology of DS and AD includes endolysosomal impairments represented by enlarged endosomes [61], which has been linked to the accumulation of C99.

Niemann-Pick disease type C (NPC) is an autosomal recessive neurodegenerative disease associated with mutations in the NPC1 or NPC2 genes of human chromosome 18, which code for lysosomal lipid transport proteins. Dysfunction of this transport protein causes the accumulation of free cholesterol in lysosomes throughout the body and accumulation of sphingolipids, especially in cranial nerve cells. Accumulation of APP-CTFs, including C99, and changes in their localization occur in model cells of genetic and drug-induced NPC [62,63], which shows similar pathology with AD such as NFT and Aβ accumulations [64,65,66,67]. Interestingly, in Purkinje cells in the brains of NPC patients with, C99, but not Aβ42, it accumulates in early endosomes and induces a redistribution of lysosomal proteins, such as cathepsin D, to the endosomes. Additionally, enlargement of Rab5-positive early endosomes has been observed in the brains of NPC patients [62], similar to AD and DS pathology. The extent of the contribution of C99 to NPC pathogenesis needs to be verified in the future. However, since NPC is a disease associated with abnormal lipid and cholesterol metabolism and accumulation, these impairments may affect C99 metabolism. Furthermore, these lines of evidence also suggest that accumulated C99 itself may be involved in disturbances of endolysosomes and lipid metabolism, creating a vicious cycle.

## 5. Structure and Metabolism of C99

To consider the unique properties of C99, a comparison with C83, would be helpful. To repeat from the previous chapter, compared to C83, C99 is challenging to produce, but difficult to break once formed. What produces the characteristic properties of C99? Here, we focus on the function of the Aβ N-terminal region (Aβ-Nt; Aβ 1–16aa), which is missing in C83. The sequence differences in this region between humans and mice are related to aggregation and exerted toxicity [68], as well as specific in APP among other family proteins such as APLP1 and APLP2 [69], and their involvement in AD pathogenesis has attracted attention [70].

Genetic evidence suggests the importance of this region. The first is the Swedish mutation, which is present just before the β1 cleavage site and increases the production of C99 and Aβ. The second is the English (H7R) and Tottori (D8N) FAD mutations, which affect the formation of Aβ fibril [71] and oligomer [72], that are deeply involved in AD pathogenesis [31]. Furthermore, the presence of the Uppsala mutation (deletion of Aβ_19F-24V), in which the extracellular domain of C99 is deleted, leaving Aβ-Nt, which is supposed to induce AD, also suggests a requirement for Aβ-Nt in the development of AD [73]. Biochemically, the Uppsala mutation is associated with increased Aβ production and aggregation, as well as C99 accumulation. Although the Uppsala mutation has few patients and may need further validation, these mutation carriers show a younger onset age than other FAD mutations in their forties. Their pathological progression is reported to be more rapid.

If this is the case, then how does Aβ-Nt affect the C99 structure? Solution NMR measurements using zwitterionic lipid bicelles and micelles have provided the structural information on the extracellular domain of C99 [74,75]. In those analyses, the N-terminal region (1–15) forms a disordered structure, and the following seven amino acid residues form an N-helix structure that binds to the surface of membrane components and binds to the transmembrane region via a flexible five amino acid loop. In contrast to transmembrane domain to N-helix structure is predicted to have cholesterol-binding properties in a cooperative manner [74], the structure of the Aβ-Nt is disordered, which might determine the properties through interactions by binding to other factors. It has also been suggested that, depending on membrane thickness conditions, Aβ-Nt adopt a β-hairpin structure lifted on membrane surface and function as a starting point for Aβ aggregation in concert with other extracellular domains [75] (Figure 4).

So how does Aβ-Nt affect C99 properties? C99 undergoes several metabolic pathways, one of which is the γ-secretase pathway. Although, Aβ production is crossly correlated with the C99 level, C99 is not a good substrate for γ-secretase compared to C83 and other γ-secretase substrates, Notch. The γ-secretase activity analysis in vitro [76] or cell-based reporters [77] indicates that C83 and Notch are cleaved more easily than C99. Two reasons are postulated: the first is the difference in the length of the extracellular region, and the second is that this extracellular region has an inhibitory function on γ-secretase cleavage. On that basis, a drosophila reporter system has shown that the shorter the extracellular domain, the more efficiently the substrate is cleaved by γ-secretase substrates [78]. Furthermore, the Aβ-Nt sequence has an inhibitory structure for γ-secretase cleavage [76], and C99 accumulation has been predicted to saturate the substrate binding site of γ-secretase, thereby decreasing its activity [79].

The other major metabolic pathway is degradation in the autolysosomal pathway. C99 and other APP-CTFs have been reported to be transported to lysosomes and subjected to degradation by β-cathepsin [80]. APP-CTF in the endocytic pathway interacts with a complex formed by AP2 (adaptor-related protein complex 2) and, one of the AD risk factors, PICALM (phosphatidylinositol clathrin assembly lymphoid-myeloid leukemia). This complex interacts with autophagy protein LC3 (microtubule-associated protein one light chain 3) to transport C99 to the autolysosomal pathway (70, 71). Autolysosomal pathways are affected by the efficiency of vesicular transport by endosomes, and disturbances in both pathways have been reported in AD and other neurodegenerative diseases [81]. As will be discussed in more detail in later chapters, accumulated C99 may contribute to the impairment of these pathways and promote a vicious cycle of AD progression.

## 6. Formation of Intracellular Pathology by C99 Accumulation

Although the possibility that C99 has some physiological activity cannot be completely ruled out, in view of its pathological accumulation in AD, DS, and NPC, it is reasonable that C99 has a pathology-specific and gain-of-function nature.

The presence of Aβ-independent toxicity is also suggested by results from BRI-Aβ transgenic mice. BRI-Aβ is a system that efficiently produces and secretes Aβ by linking the transmembrane region of the BRI2 protein associated with familial British and Danish dementia (FBD and FDD) to an Aβ sequence, and the linker sequence connecting the two is cleaved by furin [82]. Interestingly, mice expressing BRI2-Aβ42 showed age-dependent deposition of Aβ in the brain parenchyma and blood vessels [83], but no decline in cognitive function [84]. This result contrasts with the unique neurotoxicity of peptides containing the C99 structure, which was not seen with Aβ treatment [85], and the cognitive decline seen in the mouse model overexpressing C99 [86]. Of course, simple comparisons should be made with caution, as the expression systems and expressed proteins are very different, but it suggests that there are mechanisms of neurological dysfunction by APP metabolites other than Aβ, and that C99 is a strong candidate for such mechanisms. This chapter outlines the pathogenesis of C99 accumulation (Figure 5) in terms of the following: (1) endolysosomal dysfunction; (2) disruption of intracellular cholesterol transport; and (3) effects on neuronal function.

### 6.1. Endolysosomal Anomalies and C99

Intracellular vesicular trafficking is essential for the transport and degrading intracellular or phagocytosed materials through the endolysosomal or autolysosomal pathway. These pathways’ impairment is a possible neurodegeneration trigger [87,88].

Similarly, the “traffic jam hypothesis”, which focuses on vesicular traffic impairment in AD, has been proposed and may complement the amyloid hypothesis as an intracellular pathology [89]. Morphological abnormalities in endolysosomes, represented by endosomal enlargement, have been reported in AD and DS [61,90,91,92]. Furthermore, it has been shown that this may be an early manifestation that occurs before Aβ accumulation [61,92]. Intriguingly, the analysis of AD and DS begins to show that C99 accumulation is a trigger of vesicular traffic impairment.

Early endosomal enlargement is observed in DS model mice, along with increased expression of the APP gene due to its triplication [93,94,95]. Furthermore, cholinergic neuropathy occurs in this mouse model, along with retrograde axonal transport defects of NGF, suggesting a relationship to the pathology [90]. The partial knockout of APP [94] or BACE1 [95] ameliorates endolysosomal defects in DS model mice, suggesting the involvement of C99. Endosomal enlargement is observed in various model systems, such as DS patient-derived fibroblasts [96], BACE1 overexpressing model cell systems [57,97], C99 overexpressing cells [57,98], and isogenic iPS cells with the mutations of APP and PS1 [52]. In many of these cell models, pharmacological analysis has shown that endosomal pathology is ameliorated when C99 and Aβ production is suppressed with BACE1 inhibitors, but worsened when Aβ production is suppressed with γ-secretase inhibitors and C99 accumulates. These lines of evidence suggest that the mediator of vesicular traffic impairment is C99 accumulation, an Aβ-independent exertion of toxicity.

Several hypotheses have been proposed that the molecular mechanisms involving C99 accumulation leading to vesicular traffic impairment. One of them is the activation of the Rab5 protein [7,99]. The Rab GTPase family proteins are critical regulators of vesicular trafficking and are presumed to be associated with the development of many neurodegenerative diseases [100]. Rab proteins switch between GTP binding active state and GDP binding inactive state, with the active form forming a complex with accessory proteins to regulate the fusion and fission of transport vesicles [101]. Rab5 is an early endosome marker, and its abnormal activation has been involved in AD pathogenesis [99]. As aforementioned, enlarged endosomes have been observed as Rab5-positive endosomes in AD, DS, and both diseases model mouse brains. Notably, the Rab5 overexpression, especially the constitutively active mutant (GTPase-deficient) leads to endosome enlargement. This is likely since Rab5 is involved in vesicle fusion, and excessive activation leads to membrane fusion, preventing subsequent maturation of endosomes. In Rab5 transgenic mice, without Aβ and C99 accumulation, somewhat similar to AD is observed, including memory impairment, and cholinergic neuronal cell death [102]. This suggested that Rab5 activation exists downstream of C99 mediated pathology.

How does Rab5 activation occur? As a molecular mechanism, an interesting hypothesis has been proposed that APPL1, one of the accessory proteins that binds to activated Rab5, interacts with the C-terminal portion of C99, thereby forming a rigid complex with Rab5 and maintaining its activated state. The co-localization of APPL1 and Rab5 in enlarged endosomes in the brains of AD and DS patients also strengthens this hypothesis [98]. However, the binding site of APPL1 is the C-terminus of C99, a region also shared by full-length APP and C83. Although presumably due to the simultaneous localization of APPL1 and C99 to the endosomes, the role of Aβ-Nt, which may control the specific mechanism of C99, remains a mystery.

Next, we focus on the mechanism of action of the C99-specific Aβ-Nt region and consider another aspect of C99-mediated vesicular traffic impairment. In addition to Rab proteins, lipid composition, especially the localization of phosphatidylserine (PS), is essential for endosome fission and fusion [103,104]. PS showed asymmetric localization on the cytoplasmic side in the lipid bilayer of the plasma membrane/endosome. The cytoplasmic side maldistribution of PS in endosomes promotes membrane bending and regulates endosomal fission and fusion, i.e., vesicular trafficking, by recruiting PS-binding proteins, such as Evectin-2 [105,106,107]. The asymmetric PS distribution is produced by lipid flippase (LF), which translocates PS to the cytoplasmic side [108]. Most LFs are formed in a complex with TMEM30A and its active domain, the P4-ATPase family protein, and the complex formation is essential for LF activity [109,110]. In particular, LFs formed by TMEM30A and ATP8A1 localize to endosomes and regulate their function [100,104]. Interestingly, the accumulation of C99 leads to the formation of an abnormal complex with TMEM30A [97], resulting in reduced LF complex formation and activity [57]. In AD model mice, such as NL-G-F knock-in and A7 mice, the complex formation of TMEM30A and C99, LF complex formation failure, and decreased membrane localization of Evectin-2 occurred before Aβ deposition [57]. Along with decreased LF activity, Evectin-2 binding to endosomes is reduced, and endosomal enlargement is promoted, a possible mechanism that would explain the vesicular trafficking impairments caused by C99. Since functional deletion of TMEM30A in the Drosophila model leads to the accumulation of Rab proteins, including Rab5 [111], abnormal PS localization may also affect Rab5 activity, and future analysis is required.

### 6.2. Autolysosomal Anomalies and C99

Abnormalities in endosomal and vesicular trafficking are also accompanied by impaired function of the lysosomes and autophagy that follows. A previous study reported that C99 accumulates in enlarged cathepsin B and LAMP1-positive structures in 2xTg, and 3xTg mice [56]. Furthermore, the neurons with FAD mutations in PSEN1 and APP show impairment of lysosomal functions, and inhibition of γ-secretase to accumulate C99 induces lysosomal dysfunction, which is restored by BACE1 inhibition to reduce C99 [51]. Interestingly, acidification of the lysosomes is impaired in AD models, resulting in a marked accumulation of lysosome marker proteins such as LAMP1 [58]. Furthermore, similar lysosome defects occur in DS primary fibroblasts and show the dependency on C99 accumulation [112]. This evidence suggests that lysosomal abnormalities are widely implicated in AD pathogenesis and are attributable to C99 accumulation. The activity of many lysosomal proteins, such as cathepsin-D, requires low pH, produced by the proton pump V-ATPase. A recent study using DS fibroblasts suggested that lysosome accumulation and dysfunction are caused by the interaction with phosphorylated APP or C99 and the Vo subunit of V-ATPase [113]. Furthermore, this model cell system also showed that enzyme activities such as cathepsin-D decreased with increasing lysosome pH.

The disruption of this autolysosomal pathway may overturn the previous theory of the formation of senile plaques, a characteristic pathology in AD. It has been thought that Aβ secreted “extracellularly” acts as a core and attracts other Aβ, leading to senile plaque formation. However, from multiple AD model mice analysis, decreased acidity of autolysosomes impairs the proteolytic function and promotes the accumulation of Aβ aggregates within autolysosomes. These Aβ aggregate-containing lysosomes accumulate around the nucleus forming a petal-like structure, which is named “PANTHOS,” and distorts the shape of neurons and cause neuronal cell death as they engulf the plasma membrane and other intracellular organelles. This report suggests that the remaining Aβ aggregates finally act as a core to promote senile plaque formation [114]. Although C99 accumulation impairs the autolysosome function, the extent to which it contributes to the formation of PANTHOS remains to be analyzed. However, this report proposes that most pathologies of the amyloid hypothesis begin “inside” the cell. C99 accumulation, along with aggregated Aβ, may also be a factor in the formation of the inside pathology of the amyloid hypothesis, which has not been elucidated.

### 6.3. Intracellular Cholesterol Trafficking and C99

The hypothesis based on the structural properties of C99 have been proposed. As discussed in the chapter on the structure of C99, several studies have predicted that C99 binds to cholesterol [74,115], suggesting that it may be involved in cholesterol metabolism and transport.

In addition to endosomes, γ-secretase and APP accumulate in mitochondria-associated endoplasmic reticulum (ER) membranes (MAM), which are organelles that link ER and mitochondria [116]. Mitochondrial dysfunction is critical in AD pathogenesis [117], but the mechanism is still unclear. Interestingly, it has been reported that C99 accumulation is associated with impaired mitochondrial function, morphology, and mitophagy in patients with AD and model cells and animals [118]. It was hypothesized that AD pathology and decreased γ-secretase activity promote the accumulation of C99 in the MAM, resulting in reduced mitochondrial function [119], and C99 acts as a cholesterol sensor, forming a “hard” structure with the lipid bilayer, a microdomain called a detergent-resistant membrane (DRM), and accumulation of intracellular and extracellular cholesterol in the MAM [120,121]. In support of this hypothesis, decreased γ-secretase activity and APP-CTF accumulation triggered the accumulation of lipid droplets, which are intracellular accumulations of lipids such as cholesterol [122]. C99 might localize to surfactant-resistant regions in the endosomes [57], suggesting that the cholesterol-binding properties of C99 influence organelles other than the ER. The pathological contribution of cholesterol-binding properties of C99 is noteworthy since microdomain changes such as DRM, in addition to promoting Aβ production by accumulating each secretase and APP [123,124,125,126,127], also affect physiological changes in neurons [128].

On the other hand, as discussed in the previous chapter, structurally, the N-terminal site of Aβ is not essential for cholesterol binding, so how does the C99 specificity plays a role in this hypothesis focuses attention.

### 6.4. Effects of C99 Accumulation on Neuronal Function

Interestingly, the accumulation of C99 in AD patients is more pronounced in the hippocampus in CA1 and the dentate gyrus, and its accumulation level correlates with cognitive decline [54]. C99 accumulation may affect neurological function. The possible involvement of C99 accumulation in changes in neurological function as pathological changes in MCI and early AD is briefly discussed in this chapter.

In the hippocampus, two different frequencies of brain waves, theta (4–8 Hz) and gamma (26–70 Hz) waves, are activated in the association (coupling), which is related to cognition such as learning and memory. In AD and MCI, it has been reported that theta-gamma coupling is disrupted [129]. This is deduced as an initial pathology of AD. Reduced theta-gamma coupling was also observed in the TgCRND8 [130] and J20 mouse model [131] before Aβ aggregation. Interestingly, BACE1 inhibitors improve these abnormalities and correlate with a decrease of C99 in AD model mice [131]. These changes in the overall brain function are expected to be associated with the changes in the function of individual neurons.

Abnormalities in synaptic plasticity, an essential cellular process of memory and learning, are found in AD. It is well known that Aβ accumulation or oligomers can disrupt synaptic plasticity, including long-term potentiation (LTP) and long-term depression (LTD). C99 accumulation has also been reported to affect synaptic plasticity. C99 transgenic mice reduced LTP but not LTD [132]. Adeno-associated viral vector -mediated expression of C99 decreased LTP, and γ-secretase inhibitor administration worsened LTP, suggesting the decreasing effect of C99 on LTP, independent of Aβ [56].

Some potassium channels including KCNQ are associated with epilepsy [133], which is an early manifestation of AD [134]. C99 has a sequence similarity with the voltage-gated potassium channel β-subunit and interacts with KCNQ2/3 to modulate their activity [135]. Notably, the expression and activity of BACE1 is also reported to be involved with KCNQ2/3 distribution. KCNQ2/3 forms a complex with BACE1 [136], and palmitoylation of BACE1, which determines lipid raft localization [137], is essential for KCNQ2/3 localization in lipid rafts [138]. BACE1 and its product, C99, are regulators of potassium channel localization, and this hypothesis could explain the epileptiform symptoms of AD. Their relevance to AD patient pathology is noteworthy.

Thus, C99 accumulation has potential to induce abnormal neurological functions. Also, the impairments of endolysosomes, autolysosomes, and cholesterol transport described in the previous section, may be closely related to neurological function. However, the extent to which these contribute to symptoms of cognitive function in AD and MCI is not fully identified. Further mechanistic clarification, such as using specifically inhibit C99 mediated abnormalities, is required.

## 7. Problems and Future Directions

The hypotheses and molecular mechanisms left some problems that need to be resolved. First, Aβ and C99 toxicity are closely related and may be difficult to separate. Intracellular Aβ oligomers (iAβo) have toxicity [139,140], very similar to that of C99. FAD mutations, Arctic (E22G) and Osaka (Δ22) in the middle part of the Aβ sequence, promote iAβo formation in vivo [141,142]. iAβo accumulate in endosome and correlated with cognitive impairments prior to the extracellular accumulation of Aβ impaired vesicular trafficking and autolysosomal function [142,143,144]. Structurally, under certain conditions, the N-terminal portion of C99 has a β-strand structure, which is presumed to attract and oligomerize Aβ [75], suggesting that C99 accumulation and iAβo toxicity may be closely related.

Second, there is the issue of analytical methods. To distinguish between C99 and Aβ functions, secretase inhibitors and specific detection methods have been devised. However, each method has its limitations, including the risk of inhibitors affecting other secretase substrates and the tendency toward artificial and subjective analysis. To develop a simple and objective method to detect C99 toxicity, and an animal model that can better separate C99 and Aβ toxicity is required.

However, as discussed, C99 toxicity surely exists. Since it shares the same production pathway and structure with Aβ, it could be a target for multifunctional drugs that simultaneously treat both toxicities.

In the next section, we discuss BACE1 inhibitors and C99-targeting drugs that have the above potential.

### 7.1. BACE1 Targeting

Since BACE1 is one of the enzymes responsible for Aβ production and no phenotype was initially observed in BACE1 knockout mice [145,146], BACE1 has been considered as an ideal target for AD therapy. Furthermore, inhibition of BACE1 activity suppresses C99 and Aβ production that could improve intracellular and extracellular pathologies. However, targeting BACE1 also presents some challenges. First, BACE1 cleaves many substrates other than APP. Especially, the cleavage of Neuregulin1 type III (NRG1) is well analyzed. BACE1-cleaved NRG1 works as a ligand for the ErbB receptor family and promotes intercellular signaling to mediate myelin sheath formation [36,37,147] and synaptic plasticity [39]. Additionally, behavioral and memory task abnormalities in BACE1 knockout mice were gradually evident [39,40,148]. Second, the amyloidogenic processing of APP is unique among BACE1 substrates. It has been reported that amyloidogenic processing is mainly executed in the membrane microdomain. In contrast, NRG1 is a better substrate for BACE1 cleavage than APP due to its structural properties and non-raft localization. Simple BACE1 inhibitors are more effective at inhibiting NRG1 cleavage than APP [149], which may make it more challenging to avoid side effects.

APP cleavage-specific BACE1 inhibitors have been devised that the take advantage of the localization of APP and BACE1s to the lipid microdomain. Cholesterol labeled BACE1 inhibitor to target the lipid microdomain, effectively and selectively inhibits APP cleavage [149,150]. However, currently, there are many difficulties in reducing the side effects of BACE1 inhibitors. By targeting the Aβ-Nt site of APP, APP-specific BACE1 cleavage can be inhibited, which may circumvent the problems with BACE1 inhibitors mentioned above. This will be discussed in the next chapter.

### 7.2. C99 Targeting

Targeting C99 is expected to lead to the development of drugs with multifunctional properties that can ameliorate many unfavorable conditions in the amyloid hypothesis. In this chapter, we outline the actions of antibodies and peptides that bind to the Aβ N-terminus and discuss their potential.

As a first effect, the factors binding to this region can modify each secretase cleavage and reduce the production of C99 and Aβ. Treatment with an mC99_1–7_ antibody, which recognizes the Aβ 1–7 region with HEK293 cells stably expressing the Sweden mutant of APP, can be internalized in the cell, detect intracellular accumulated full-length-APP/C99, and inhibit Aβ production [151]. Furthermore, treating this antibody in the AD model mice (APPPS1) reduced soluble Aβ. The intrabody expression targeting Aβ 3–6 also decreased Aβ production [152]. One protein that binds to C99 and full-length APP via Aβ-Nt is integral membrane protein 2B (ITM2b/BRI2), the gene responsible for familial British and Danish dementia (FBD and FDD, respectively). FBD and FDD mutations promote the amyloid fibrils, containing partial BRI2 domains called ABri and ADan. On the other hand, the function of APP, which binds to BRI2, is presumed to play a significant role in the mechanism of cognitive decline [153]. BRI2 binds to APP and C99 through the Aβ N-terminal region and ceases secretase cleavages by α, β, and γ secretases. A series of peptides, termed MoBA (modulator of β-cleavage of APP), have been identified as APP binding sites of BRI2. Interestingly, β-secretase inhibitors and MoBA, especially N3-2A (DAVYYCGIKY), strongly suppressed C99 production and rescued LTP deficiencies in FBD KI mice [154], suggesting that MoBA has the potential as an APP-specific BACE1 inhibitor. C99-binding peptides have also been reported to inhibit γ-secretase cleavage of C99. A group of peptides, which have CDCYCxxxxCxCxSC motifs, especially #4 (MHLVICDCYCTTDICYCYSCTPN), bind to Aβ-Nt of C99 [76]. They showed that #4 and WO2, 82E1, and 6E10, antibodies that bind to Aβ-Nt, inhibit Aβ production by in vitro γ-secretase assay using C99 as a substrate. Their results suggest that the ectodomain structure of C99 is essential for γ-secretase recognition. Inhibition of Notch cleavage has been an obstacle to developing γ-secretase inhibitors for treating AD since they induce side effects; however, #4 and these antibodies did not affect the Notch cleavage activity. This peptide can bind to APP and inhibit β-secretase cleavage without affecting the cleavage of the other BACE1 substrate, sialyltransferase one. Since MoBA binds to full-length APP but not to C99 [154], the differences in their respective structures must be considered. However, the analysis of MoBA and C4 suggests that binding of the molecules to full-length APP and C99 via Aβ-Nt leads to specific β-secretase inhibition and specific γ-secretase inhibition, respectively.

As a second effect, factors that bind to this region may ameliorate Aβ aggregation and toxicity: in contrast to the C-terminal portion of Aβ, which is hydrophobic and forms β-sheet structures in Aβ aggregates, Aβ-Nt is relatively hydrophilic and unstructured [70,75,155,156] and can be exposed on the surface of oligomers and fibers [157], which is thought to regulate toxicity [158]. It has also been reported that Aβ-Nt contains binding sites for heavy metals such as copper [159,160] and zinc [161,162], which contribute to AD pathogenesis and are closely associated with Aβ toxicity. The epitopes of antibodies that specifically recognize oligomers and protofibrils overlap with Aβ-Nt [163], and targeting this site may improve toxicity against these aggregates. On the other hand, Aβ-Nt is also involved in aggregation, and 3D6 (Aβ1–5), which has an epitope at the same site, can inhibit Aβ aggregation in vitro [164] and in vivo model [42]. These results suggest that targeting Aβ-Nt may help ameliorate the toxicity of Aβ aggregation, but on the other hand, antibodies that recognize Aβ deposited in blood vessels have been associated with a risk of intracerebral hemorrhage [165], and there may be risks in simply targeting this site. Whether peptides or compounds that bind to this site can overcome the risk posed by antibodies may need to be studied with caution.

As a third effect, those that bind to this region may also be able to ameliorate C99’s unique intracellular pathologies. Our group found that the binding of TMEM30A to C99 in endosomes prevents the binding of P4-ATPase, the original TMEM30A partner molecule ATP8A1, and reduces lipid flippase activity, which may lead to endolysosomal damage [57,97]. We explored the C99-binding region of TMEM30A and found a 25 amino acid sequence named T-RAP (Tmem30A-related amyloid-beta interacting peptide). Interestingly, T-RAP, which binds to full-length APP and C99 via the Aβ-Nt, reduced the accumulation of C99. Furthermore, in a cultured cell AD model system with a constant expression of BACE1, reduced LF complex formation and activity and endosomal hypertrophy was ameliorated by the T-RAP treatment [57]. As with the peptides with similar properties described, decreased C99 production from full-length APP may be involved. However, since T-RAP can also suppress decreased LF activity when C99 is expressed alone, we proposed the hypothesis that T-RAP inhibited the abnormal complex formation between C99 and TMEM30A. While we need further approach to probe this hypothesis, targeting Aβ-Nt promisingly ameliorates intracellular dysfunctions by C99 accumulation.

Targeting the Aβ-Nt in this way may allow for multifunctionality (Figure 6), leading to the control of Aβ production and aggregation, which has been primarily studied in the amyloid hypothesis, thereby reducing the overlooked intracellular toxicity of C99.

## 8. Conclusions

The efficacy of Aβ immunotherapies has been demonstrated, and their development for AD is starting a new era. However, several studies have demonstrated a variety of intracellular pathologies independent of the toxicity of extracellular Aβ aggregates in the amyloid hypothesis. Although not discussed in this review, tau accumulation and chronic inflammation are major pathological features in AD. AD can be considered as a complex disease with a mixture of various pathologies. In the future, Aβ antibody therapy will be the mainstay therapy, and multidrug combinations for toxicity independent of Aβ will be devised. A re-evaluation of secretase inhibitors, BACE1 inhibitors, and γ-secretase modulators that spare Notch cleavage [33], would be performed. However, as patients with AD have underlying diseases and are taking multiple drugs, developing a drug with multiple therapeutic effects is desirable. Therefore, drugs targeting C99 may ameliorate Aβ-independent intracellular impairments and inhibit Aβ production and aggregation and are expected to be multifunctional drug targets to comprehensively treat amyloid hypothesis-based toxicity.

## Figures and Tables

**Figure 1 cells-12-00454-f001:**
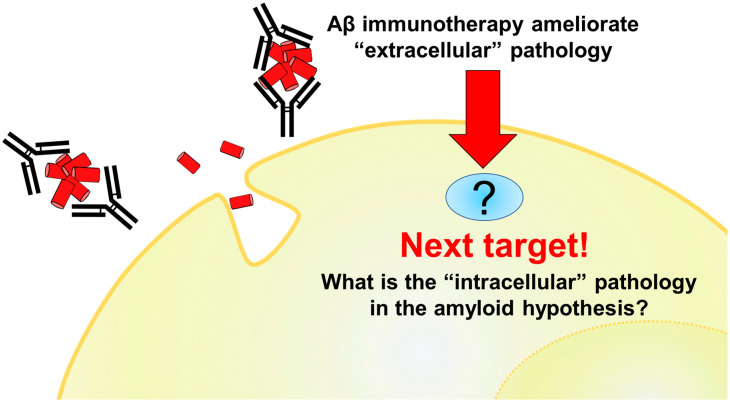
A diagram of the subject of this review. The therapeutic efficacy of antibodies against extracellularly aggregating Aβ has been demonstrated and is expected to be the first step toward curative treatment of AD. On the other hand, their therapeutic effects are still limited, and it will be necessary to develop treatments from a different perspective. This review outlines the hidden aspect of the amyloid hypothesis: intracellular pathogenesis and its potential as a therapeutic target.

**Figure 2 cells-12-00454-f002:**
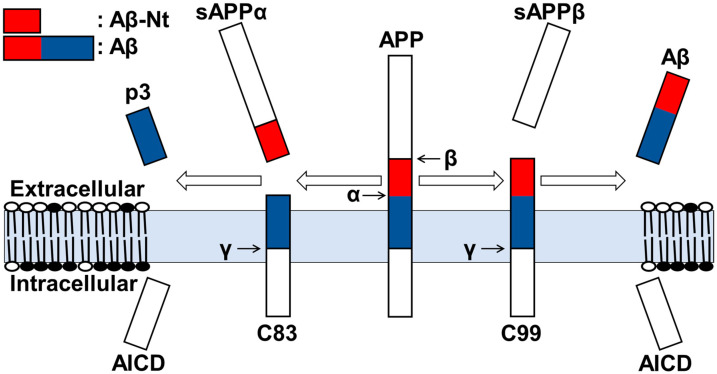
Processing by each secretase of APP. Shedding of the extracellular domain is carried out by α-secretase or β-secretase to produce C83, sAPPα, or C99, sAPPβ, respectively; intracellular sequences of C83 and C99 are cleaved by γ-secretase to produce P3 and Aβ, respectively. The Aβ N-terminal region (Aβ-Nt) of interest in this review is indicated in red, and the Aβ region is indicated in red plus blue.

**Figure 3 cells-12-00454-f003:**
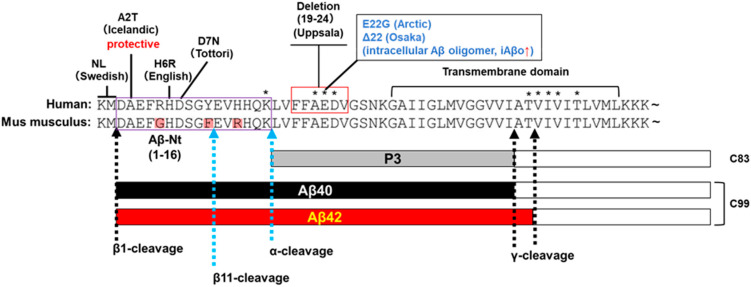
A schematic view of the C99 sequence. Mus musculus sequence comparison is included to discuss differences in Aβ-Nt (purple square). The names and sites of the FAD gene mutations described in this review are listed and mutations associated with increased intracellular Aβ oligomers (iAβo) are indicated in blue. Deletion sites for the Uppsala mutation are indicated by red squares. Other mutations within the Aβ sequence that are not mentioned are marked with an asterisk (*). The difference between Aβ40 and Aβ42 is due to fluctuations in the γ-secretase cleavage site.

**Figure 4 cells-12-00454-f004:**
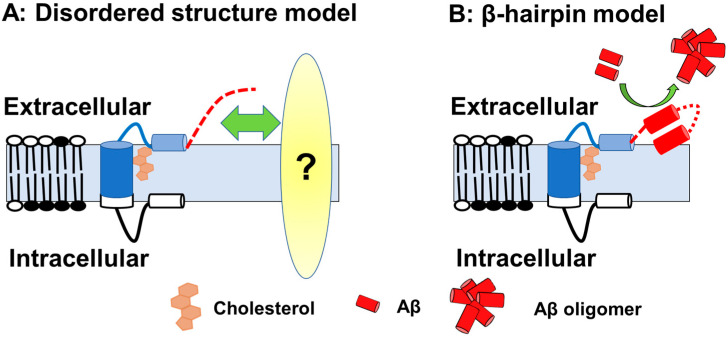
A schematic of the C99 structure. (**A**) Model with Aβ-Nt as the disordered structure; Aβ-Nt may bind to its partner molecule and exert specific functions. Red dotted line represented N-terminal structure. (**B**) Model assuming Aβ-Nt forms a β-hairpin. The β-hairpin structure formed may promote Aβ aggregation.

**Figure 5 cells-12-00454-f005:**
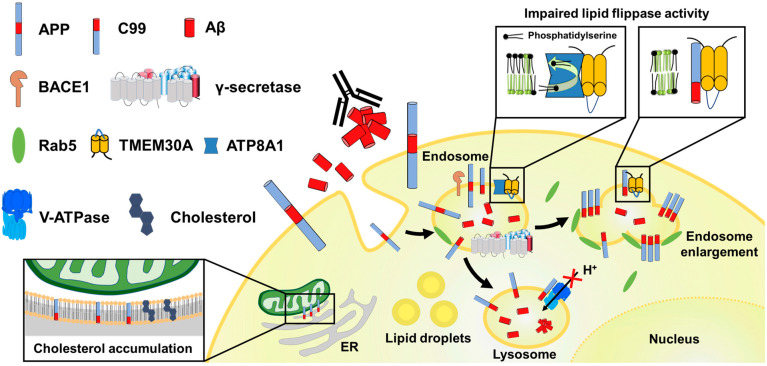
Diagram showing the intracellular pathology caused by C99 accumulation. Endolysosomal damage due to abnormal activation of Rab5 and decreased lipid flippase activity, autolysosomal damage due to decreased V-ATPase activity, and cholesterol intracellular transport defects induced by C99 accumulation are illustrated.

**Figure 6 cells-12-00454-f006:**
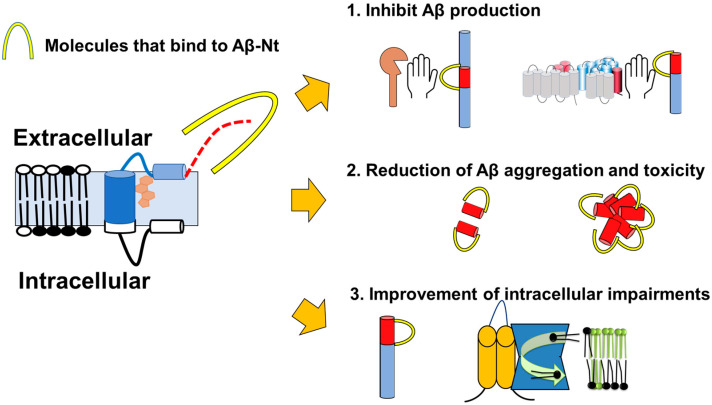
Predicted multifunctionality of Aβ-Nt-binding C99-targeted drugs. They may (1) inhibit APP cleavage by BACE1 and C99 cleavage by γ-secretase, (2) inhibit Aβ aggregation and reduce the toxicity of aggregates, (3) inhibit C99 binding to partner factors and reduce intracellular pathology (lipid flippase is shown as an example).

## Data Availability

Not applicable.

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
