# Peer review of "The Pursuit of the “Inside” of the Amyloid Hypothesis—Is C99 a Promising Therapeutic Target for Alzheimer’s Disease?"

_cells, 2023, doi:10.3390/cells12030454_

Round 1
Reviewer 1 Report
In this review, the authors summarized findings from various studies on Alzheimer's disease (AD) pathogenesis and insisted thier brand-new hypothesis that C99 would be a new therapeutic target for AD. Since several studies showed that intracellular accumulation of pathological proteins could cause neurotoxicity, thier new hypothesis sounds reasonable. They also carefully discussed on pros and cons of their new hypothsis to avoid misleading readers. Taken together, this review paper would be stimulating and helpful for many readers who are working on AD research.
Author Response
Thank you for taking the time to review this review. We are relieved to receive a good evaluation. Since no corrections seem to have been pointed out here, we will proceed.
Reviewer 2 Report
This review is well written. Figures are fine.
They discussed whether APP C-terminal fragment(CTF) is a target for AD instead of Abeta. Few researchers think about APP CTF as a target for AD.
The authors could cite the following article to strengthen their C99 hypothesis.
Neurotoxicity of a carboxyl-terminal fragment of the Alzheimer's amyloid precursor protein.
Kim SH, Suh YH.J Neurochem. 1996 Sep;67(3):1172-82. doi: 10.1046/j.1471-4159.1996.67031172.x.PMID: 8752124
Author Response
Thank you for taking time out of your busy schedule to peer review and provide valuable feedback.
Following your advice, we have cited the literature in the following context.
[85] is the relevant citation.
****
The presence of Aβ-independent toxicity is also suggested by results from BRI-Aβ transgenic mice. BRI-Aβ is a system that efficiently produces and secretes Aβ by linking the transmembrane region of the BRI2 protein associated with familial British and Danish dementia (FBD and FDD) to an Aβ sequence, and the linker sequence connecting the two is cleaved by furin [82]. Interestingly, mice expressing BRI2-Aβ42 showed age-dependent deposition of Aβ in the brain parenchyma and blood vessels [83], but no decline in cognitive function [84]. This result contrasts with the unique neurotoxicity of peptides containing the C99 structure, which was not seen with Aβ treatment [85], and the cognitive decline seen in the mouse model overexpressing C99 [86].
Reviewer 3 Report
This article represents a high-quality contribution to the relatively little-discussed role of the C99 fragment in AD pathology. I only suggest minor changes regarding capital letters and better text clarity. Although I am not able to evaluate the quality of English, I would recommend going through the text and improving some rather difficult expressions.
42: „small protein called amyloid“ , but amyloid is a peptide, not a protein
55-56: Here and in several other places, capital letters are strangely and inconsistently given, the same with Presenilin and presenilin.
157: „such as iPS and iPSC cells“. However, induced pluripotent stem cells = iPS cells or iPSCs
157: “Note that similar accumulation of C99 is also reported in a cellular AD model, such as iPS and iPSC cells artificially introduced familial AD mutations of APP and Presenilin1” – this does not make sense (at least to me)
376: V-ATPase, not v-ATPase. Not V0, but Vo (as oligomycin-inhibited)
400: “In addition to endosomes, γ-secretase and its substrate APP accumulate in mitochondria-associated endoplasmic reticulum (ER) membranes (MAM)” … as the authors also mention, APP is not a substrate of γ-secretase
480: “psesses”
544-549: I thing that better formulation would improve the understanding of the paragraph
Author Response
Thank you for taking time out of your busy schedule to peer review and provide valuable feedback.
All errors you have pointed out have been corrected in the revision.
The sections that were pointed out to be difficult to understand have been shortened by dividing the subject into cohesion and toxicity. They are as follows.
In some parts of the report, only the review literature was cited, so we have added the original literature.
***
As a second effect, factors that bind to this region may ameliorate Aβ aggregation and toxicity: in contrast to the C-terminal portion of Aβ, which is hydrophobic and forms β-sheet structures in Aβ aggregates, Aβ-Nt is relatively hydrophilic and unstructured [70,75,155,156] and can be exposed on the surface of oligomers and fibers [157], which is thought to regulate toxicity [158]. It has also been reported that Aβ-Nt contains binding sites for heavy metals such as copper [159,160] and zinc [161,162], which contribute to AD pathogenesis and are closely associated with Aβ toxicity. The epitopes of antibodies that specifically recognize oligomers and protofibrils overlap with Aβ-Nt [163], and targeting this site may improve toxicity against these aggregates. On the other hand, Aβ-Nt is also involved in aggregation, and 3D6 (Aβ1-5), which has an epitope at the same site, can inhibit Aβ aggregation in vitro [164] and in vivo model [42]. These results suggest that targeting Aβ-Nt may help ameliorate the toxicity of Aβ aggregation, but on the other hand, antibodies that recognize Aβ deposited in blood vessels have been associated with a risk of intracerebral hemorrhage [165], and there may be risks in simply targeting this site. Whether peptides or compounds that bind to this site can overcome the risk posed by antibodies may need to be studied with caution.
Reviewer 4 Report
Takasugi and co-workers present in this manuscript an overview on C99 as a potential therapeutic target against AD. This is well written and presented in a clear and concise way. It should be appropriate for publication in cells. This manuscript could be further improved by addressing the following comments.
1. Abstract should be updated to include Lecanemab (LEQENBI) very recently approved by FDA.
2. A reference for amyloid hypothesis may be incorrect. Cite the original paper by Hardy and Higgins.
Hardy JA, Higgins GA. Alzheimer's disease: the amyloid cascade hypothesis. Science. 1992 Apr 10;256(5054):184-5. doi: 10.1126/science.1566067. PMID: 1566067.
3. Kim and co-workers reported no phenotype in cognition of BRI-Abeta mice. This also supports the idea of C99 impact on AD pathogenesis.
Kim J, Chakrabarty P, Hanna A, March A, Dickson DW, Borchelt DR, Golde T, Janus C. Normal cognition in transgenic BRI2-Aβ mice. Mol Neurodegener. 2013 May 12;8:15. doi: 10.1186/1750-1326-8-15. PMID: 23663320; PMCID: PMC3658944.
4. Overall, this manuscript cites substantial review articles in addition to original research articles. It is alright if this manuscript is a research article. However, this is a review article. This reviewer would cite original research articles to respect them.
Author Response
Thank you for taking the time to peer review and provide valuable feedback.
- we have added the description as you requested.
2. Thank you for your valuable suggestions. I have added the citation you indicated and made minor changes to the text.
3.We have added the following sentence to the text based on the quote you pointed out, thank you for your important remarks.
4. You are absolutely right. We will not go into details as there are many, but we have checked the original references and added them as appropriate. On the other hand, we believe that it is also useful to look at the REVIEW to understand the current status of the highlighted sections, and we have made changes in the form of parallel references. Some of the hypotheses and theories were suitable for a review, so we have left some of them as they are.
****
As a second effect, factors that bind to this region may ameliorate Aβ aggregation and toxicity: in contrast to the C-terminal portion of Aβ, which is hydrophobic and forms β-sheet structures in Aβ aggregates, Aβ-Nt is relatively hydrophilic and unstructured [70,75,155,156] and can be exposed on the surface of oligomers and fibers [157], which is thought to regulate toxicity [158]. It has also been reported that Aβ-Nt contains binding sites for heavy metals such as copper [159,160] and zinc [161,162], which contribute to AD pathogenesis and are closely associated with Aβ toxicity. The epitopes of antibodies that specifically recognize oligomers and protofibrils overlap with Aβ-Nt [163], and targeting this site may improve toxicity against these aggregates. On the other hand, Aβ-Nt is also involved in aggregation, and 3D6 (Aβ1-5), which has an epitope at the same site, can inhibit Aβ aggregation in vitro [164] and in vivo model [42]. These results suggest that targeting Aβ-Nt may help ameliorate the toxicity of Aβ aggregation, but on the other hand, antibodies that recognize Aβ deposited in blood vessels have been associated with a risk of intracerebral hemorrhage [165], and there may be risks in simply targeting this site. Whether peptides or compounds that bind to this site can overcome the risk posed by antibodies may need to be studied with caution.